# A Synonymous Exonic Splice Silencer Variant in IRF6 as a Novel and Cryptic Cause of Non-Syndromic Cleft Lip and Palate

**DOI:** 10.3390/genes11080903

**Published:** 2020-08-07

**Authors:** Beau Sylvester, Frederick Brindopke, Akiko Suzuki, Melissa Giron, Allyn Auslander, Richard L. Maas, Becky Tsai, Hanlin Gao, William Magee, Timothy C. Cox, Pedro A. Sanchez-Lara

**Affiliations:** 1Division of Plastic and Maxillofacial Surgery, Children’s Hospital Los Angeles, Los Angeles, CA 90027, USA; beausylvester@wustl.edu (B.S.); aauslander@chla.usc.edu (A.A.); WMagee@chla.usc.edu (W.M.III); 2Operation Smile Global Headquarters, Virginia Beach, WV 23453, USA; Freddy.Brindopke@operationsmile.org; 3Department of Oral & Craniofacial Sciences, University of Missouri-Kansas City School of Dentistry, Kansas City, MO 64108, USA; asg5p@umkc.edu (A.S.); coxtc@umkc.edu (T.C.C.); 4Operación Sonrisa Honduras, Tegucigalpa 11101, Honduras; Melissa.Giron@operationsmile.org; 5Department of Preventive Medicine, Keck School of Medicine of the University of Southern California, Los Angeles, CA 90033, USA; 6Division of Genetics, Department of Medicine, Brigham and Women’s Hospital, Harvard Medical School, Boston, MA 02115, USA; RMAAS@bwh.harvard.edu; 7Fulgent Genetics, Temple City, CA 91780, USA; btsai@fulgentgenetics.com (B.T.); harrygao@fulgentgenetics.com (H.G.); 8Department of Pediatrics, University of Missouri-Kansas City School of Medicine, Kansas City, MO 64108, USA; 9Department of Pediatrics, Cedars-Sinai Medical Center, David Geffen School of Medicine at UCLA, Los Angeles, CA 90048, USA

**Keywords:** non-syndromic cleft lip with or without palate, *IRF6*, Van der Woude syndrome, synonymous, alternative splicing, splicing enhancer, splicing silencer, cryptic splice site, silent substitution

## Abstract

Missense, nonsense, splice site and regulatory region variants in interferon regulatory factor 6 (*IRF6*) have been shown to contribute to both syndromic and non-syndromic forms of cleft lip and/or palate (CL/P). We report the diagnostic evaluation of a complex multigeneration family of Honduran ancestry with a pedigree structure consistent with autosomal-dominant inheritance with both incomplete penetrance and variable expressivity. The proband’s grandmother bore children with two partners and CL/P segregates on both sides of each lineage. Through whole-exome sequencing of five members of the family, we identified a single shared synonymous variant, located in the middle of exon 7 of *IRF6* (p.Ser307Ser; g.209963979 G>A; c.921C>T). The variant was shown to segregate in the seven affected individuals and through three unaffected obligate carriers, spanning both sides of this pedigree. This variant is very rare, only being found in three (all of Latino ancestry) of 251,352 alleles in the gnomAD database. While the variant did not create a splice acceptor/donor site, in silico analysis predicted it to impact an exonic splice silencer element and the binding of major splice regulatory factors. In vitro splice assays supported this by revealing multiple abnormal splicing events, estimated to impact >60% of allelic transcripts. Sequencing of the alternate splice products demonstrated the unmasking of a cryptic splice site six nucleotides 5′ of the variant, as well as variable utilization of cryptic splice sites in intron 6. The ectopic expression of different splice regulatory proteins altered the proportion of abnormal splicing events seen in the splice assay, although the alteration was dependent on the splice factor. Importantly, each alternatively spliced mRNA is predicted to result in a frame shift and prematurely truncated IRF6 protein. This is the first study to identify a synonymous variant as a likely cause of NS-CL/P and highlights the care that should be taken by laboratories when considering and interpreting variants.

## 1. Introduction

Clefts of the lip and/or palate (CL/P) are among the most common and recognizable inborn errors of development, presenting in one in 700 live births [1]. Orofacial clefting often occurs as a component of various genetic syndromes and sequences, but approximately 70% of CL/P cases arise in apparent isolation [1]. These individuals with non-syndromic CL/P (NS-CL/P) display complex epidemiology, varying in prevalence across geographic regions and ethnic groups, and complex etiology, with numerous genetic and environmental risk factors contributing to disease [1].

CL/P and isolated cleft palate have traditionally been treated as etiologically independent [2]. However, families in which some individuals present with cleft lip and others with cleft palate only have been noted with shared mutations in a handful of genes, such as interferon regulatory factor 6 (*IRF6*) [3], tumor protein p63 (*TP63*) [4,5], fibroblast growth factor receptor 1 (*FGFR1*) [6], and msh homeobox 1 (*MSX1*) [7]. Furthermore, studies of large, multiplex families with cases of NS-CL/P have demonstrated that there are likely sub-clinical phenotypes in otherwise “unaffected” relatives, suggesting that the spectrum of disease may be broader than previously recognized [8,9,10,11].

The identification of gene variants underlying NS-CL/P has mainly been accomplished through studies of individuals, families, and larger population samples with NS-CL/P using genome-wide association studies (GWASs), linkage analysis, and whole genome and exome sequencing. Additional gene candidates have been identified through developmental and gene expression analyses in animal models [1] and inference from individual syndromes and genes disrupted in structural chromosomal anomalies [12]. The best example of this is the *IRF6* gene, which was first identified as the cause of Van der Woude syndrome [3], a disorder defined by the co-presentation of CL/P and lower lip pits, and later shown to contribute to NS-CL/P in multiple family-based studies and GWASs [13,14,15,16,17]. Animal model studies have demonstrated that a deficiency of IRF6 is associated with defects in epithelial adhesion and differentiation [18,19,20,21,22], providing potential clues for disease pathophysiology.

In this study, we performed whole exome sequencing (WES) on several family members from a large, multiplex Honduran family with NS-CL/P, identifying a novel segregating synonymous variant in *IRF6*. We demonstrate that this synonymous variant impacts an exonic splice silencer element that results in the utilization of multiple cryptic exonic and intronic splice sites, each of which is predicted to result in a frame shift and loss of function of IRF6. Although similar types of mutations have been reported in other diseases, this represents the first example underlying the presentation of CL/P.

## 2. Materials and Methods

### 2.1. Editorial Policies and Ethical Considerations

Ethical approval for this study was obtained from the Institutional Review Board of Children’s Hospital Los Angeles under CCI-08-00150. Informed consent was obtained from all participating human subjects prior to their inclusion in the study. This study was performed in compliance with the Declaration of Helsinki and the U.S. Federal Policy for the Protection of Human Subjects.

### 2.2. Patient Recruitment and Clinical Evaluation

The proband (individual III-6; Figure 1) was first identified through Operación Sonrisa (Operation Smile) Honduras during regular pre-surgical programmatic activities for screening children with cleft lip and palate. Operación Sonrisa was established in 1997 and is a country-based foundation of the international NGO Operation Smile, which has been providing free cleft care globally for almost four decades. Upon recognition that this family included multiple affected individuals across several generations, the organization flagged the family as a strong candidate for research. A three-generation pedigree was obtained, individuals were consented for research and phenotypically screened for other comorbidities or subtle signs of an underlying syndrome.

### 2.3. Samples and DNA Extraction

Saliva samples were collected from available family members using Oragene DNA Genotek 500 adult and 575 saliva collection kits. Genomic DNA was extracted from 400 μL peripheral blood using a QIAamp DNA Mini Kit (QIAGEN, Düsseldorf, Germany) following the manufacturer’s instructions. The quality and quantity of the DNA samples were assessed using a NanoDrop 8000 (Thermo Scientific™, Waltham, MA, USA) and agarose gel electrophoresis.

### 2.4. Whole Exome Sequencing

Genomic DNA libraries were prepared using TruSight One (Illumina Inc., San Diego, CA, USA) and targets were captured using a custom probe set mixture based on TruSight One. Massively parallel DNA sequencing was initially performed on five individuals (I-2, II-1, II-2, III-3, III-6) using HiSeq 4000 instruments and chemistry (Illumina Inc.). Sequence reads were aligned to the human genome using Novoalign (Novocraft, Selangor, Malaysia). Single nucleotide variants and small indel variants were called using VarScan2 and annotated using a combination of in-house tools and Alamut Batch (Interactive Biosciences division of Sophia Genetics).

### 2.5. Mutation Screening and Sanger Confirmation

Based on the family history and pedigree structure, autosomal dominant Mendelian inheritance with incomplete penetrance was considered the best model and variants shared among affected individuals were assessed with greater scrutiny. Variants with a sequencing depth of coverage <10x or genotype quality (Q < 20) were excluded from analysis. Only rare variants with a minor allele frequency of <1% in the 1000 Genomes Project (www.1000genomes.org) or Exome Sequencing Project (ESP) (esp.gs.washington.edu/drupal/) reference populations were included for analysis. Each variant was evaluated for the potential to contribute to the orofacial cleft presentation based on a single-gene Mendelian model. The literature was reviewed for individual variants and candidate genes with potential significant relationships to the phenotype and possible deleterious effects on craniofacial structures.

Sanger sequencing with forward (5′-GCCTCCTGCCACCACT-3′) and reverse (5′-GGCTAGATTCCCTAGACCA-3′) primers designed using Primer5 ver. 0.4.0 was applied to the candidate variant. The Sanger sequencing product was separated using an ABI 3730 genetic analyzer, and the data were analyzed using Chromas V1.0.0.1.

### 2.6. In Silico Analysis

The online software Human Splicing Finder (version 3.1; https://www.genomnis.com/the-system-1) [23] was used to analyze the genomic sequence in and around exon 7 of human *IRF6*. Specifically, this tool serves two functions: (1) to identify potential exonic and intronic splicing enhancer sequences, silencer sequences and cryptic splice donor and acceptor sites in a given sequence in part based on the similarity to the respective consensus binding sites of major splice regulatory factors; and (2) determine the possible impact on any such sequences by a given variant. Only those sites above the default thresholds of the software were considered as potentially relevant.

### 2.7. Minigene Splice Assay

A fragment of the human *IRF6* gene, comprising exon 7 (of nine) and 309 bp of intron 6 and 428 bp of intron 7, was amplified by polymerase chain reaction (PCR) using HiFi HotStart DNA Polymerase (Kapa Biosystems) and human genomic DNA isolated from an unaffected control individual with the following primer pair containing restriction site linkers: hIRF6-I6 forward (5′-CAATCTCGAGCCGACTCAGTCAG TATAGCGTGGG-3′; XhoI site underlined) and hIRF6-I7 reverse (5′-AGCTCTAGATGTAACTCCCTTC TTTGTTGCC-3′; XbaI site underlined). The purified product was double digested with XhoI and XbaI and cloned into a similarly digested pET01 exon trap vector (MoBiTec, Goettingen, Germany) to generate the wildtype (WT; c.921C) construct.

The c.921C>T variant was introduced by overlap PCR with high fidelity polymerase and the following primer sets: hIRF6-I6 forward and hIRF6-synon reverse (5′-CATGACCACTGACCTCCAGGATCAG-3′), and hIRF6-synon forward (5′-GGAGGTCAGTGGTCATGCCATTTATG-3′) and hIRF-I7 reverse. All constructs were verified by sequencing.Wildtype and mutant vectors were transfected into COS7 cells (CRL-1651, ATCC) by Lipofectamine 3000 (Thermo Fisher Scientific) according to the manufacturer’s protocol (n = 3 per group). Total RNA was extracted two days post transfection using QIAshredder and an RNeasy Micro Kit (Qiagen) and then cDNA was synthesized using Superscript IV (Thermo Fisher Scientific). To detect splicing patterns, PCR was carried out using pET01-ETPR04 forward (5′-GGATTCTTCTACACACCC-3′) and pET01-ETPR05 reverse (5′-GTTGACCTCGACCCACCT-3′). As a control for DNA contamination, PCR was also performed using the RNA as a template. An aliquot of each PCR was separated on a 1.2% agarose gel, stained with ethidium bromide, and the gel image capture on an C400 imager (Azure Biosystems, Dublin, CA, USA) for the quantification of individual bands. Quantification was performed in the linear range using the Azure software and manually adjusted for band size. The remainder of the amplification reaction was run on a preparative 1.2% agarose gel and the individual products were extracted using a Qiagen gel extraction kit (Qiagen) and then Sanger sequenced (Genewiz, South Plainfield, NJ, USA) using the amplifying primers.

Plasmids for the expression of Splice Regulatory Factor 1 (pcDNA-FLAG-SF2; Addgene 99021) and Splice Regulatory Factor 2 (pcDNA3.1-SC35-cMyc; Addgene 44721) were separately co-transfected with the empty pET01 vector, or either the WT (c.921C) or mutant (c.921T) vectors into COS7 cells and total RNA was extracted two days post transfection (n = 6 per group), as described above. Following cDNA synthesis, PCR products were amplified as previously described, separated on a 1.2% agarose gel and quantified as above.

## 3. Results

### 3.1. Clinical Findings

The pedigree of the family identified by Operación Sonrisa Honduras contained seven affected individuals and two unaffected obligate carriers, as well as the grandmother (individual I-2) who bore offspring with two separate partners, with clefting segregating on both sides of the pedigree (Figure 1). The proband (individual III-6) presented with unilateral cleft lip and cleft palate as did one of his paternal uncles (individual II-3). An additional paternal uncle (individual II-10) and a cousin (individual III-11) presented with bilateral cleft lip and cleft palate, while two additional cousins (individuals III-7 and III-12) and a sibling (III-3) presented with cleft palate only. No family member exhibited lip pits or other clinical signs suggestive of a syndrome. Strikingly, all overtly affected members of the family are males.

### 3.2. Exome Sequencing and Variant Analysis

Exome sequencing was performed on DNA samples from individuals I-2, II-1, II-2, III-3, and III-6, and the data from each individual were filtered for quality and coverage and then selected for variant allele frequencies of <1%. Under a dominant inheritance model, a total of 411 variants were found to be shared between affected individuals III-6, III-3, and the presumed obligate carriers II-1 and I-2. To further narrow down candidates, we applied the machine learning algorithm, DOMINO, to select only those variants in genes determined to be most likely to result in a dominant condition [24] and then focused on only those variants deemed to be very rare (i.e., minor allele frequencies of <1 × 10^−4^). This analysis narrowed the number of shared gene variants to 27 (Table 1), which included 13 intronic variants (between 3 and 20 bp from a splice site), three variants located in 3′UTRs, one six-base pair in-frame duplication, four synonymous changes, and six missense variants. The six-base pair duplication (in *SPEN*) was subsequently determined to be a much more common variant but had been bypassed in the commercial bioinformatic pipeline because it is located in a low complexity region and thus was excluded from further consideration. Of the 26 remaining variants, fourteen of the genes have previously been implicated in diseases. The genes associated with ten of these diseases were excluded as the diseases involved principally non-craniofacial tissues. Three of the disease genes are associated with variable facial dysmorphism as part of a broader spectrum of defining features and thus were deemed less likely to play a role in our family. The remaining disease gene was *IRF6*, the most prominent CL/P gene and thus an excellent candidate for the family’s presentation. After exclusion of the 13 genes implicated in other diseases, as well as the remaining variants located in introns and untranslated regions, we were left with seven initial candidate genes: *SYT11*, *KIAA1468*, *ILF3*, *IRF6, PDE7A*, *MCL1*, and *ITGB1*. The variant in KIAA1468 was a single nucleotide duplication in a splice site and was deemed unlikely to be pathogenic. The missense variant in *ILF3* (encoding interleukin enhancer binding factor 3) alters one of a stretch of five glycine residues that is positioned within the undefined C-terminus of the protein. As the affected glycine was one of 82 glycines in the final 285 amino acids of the protein (i.e., a very glycine-rich region), we deemed this variant unlikely to be causal. The missense variant in *ITGB1* is located in a rarely utilized exon which harbors a relatively high proportion of variants compared to other exons and so was excluded. The variants in the remaining four genes, *SYT11, PDE7A, MCL1*, and *IRF6*, were synonymous changes. *SYT11* encodes synaptotagmin X1, a protein involved in synaptic vesicle regulation. This gene is considered a candidate for schizophrenia. *PDE7A* encodes a high-affinity cAMP-specific phosphodiesterase that has been shown to be important for heart, muscle, and T-cell function. *MCL1* encodes a regulator of apoptosis that is a target of an approved drug treatment for myeloid cell leukemia. The remaining synonymous variant in *IRF6*, the most prominent CL/P gene, was therefore deemed the top candidate. The *IRF6* variant also had the highest scaled combined annotation dependent depletion (CADD) score of the four synonymous changes: 14.48 (https://cadd.gs.washington.edu/) [25].

The synonymous *IRF6* variant is located in the middle of exon 7 (p.Ser307Ser; g.209963979G>A; c.921C>T) and is only found in three of 251,352 alleles in gnomAD (minor allele frequency: 1.19 × 10^−5^), with all three of Latino ancestry. Importantly, Sanger sequencing validated the presence of the *IRF6* variant and confirmed segregation in an additional affected individual (III-7), thus confirming the segregation of the variant with the phenotype across both sides of the pedigree. Unfortunately, DNA samples were not available from any of the other affected members of the family.

### 3.3. Functional Testing of the IRF6 Variant

Initial visual inspection of the synonymous *IRF6* variant suggested it did not create a splice acceptor/donor site. However, in silico analyses using the Human Splicing Finder online tool, which uses multiple distinct algorithms, revealed that three separate algorithms predicted the variant would create a new exonic splice silencer (ESS) element (Figure 2A), thus potentially disrupting normal mRNA splicing.

Using an established in vitro minigene splice assay supported the prediction model of disrupted mRNA splicing. Specifically, four distinct splicing events were detected when the variant minigene construct was transfected into COS7 cells, one of which corresponded to the product size seen when the wildtype minigene construct was used (Figure 2(Bi)). The fluorescence intensity of each PCR product was determined using an Azure Biosystems C400 gel imaging system and was then adjusted for band size, which estimated the alternate products to constitute approximately 60% of the allelic transcripts (Figure 2(Bii)).

The direct sequencing of each band following gel purification revealed the nature of each abnormal splicing event and confirmed that the upper band represented a correctly spliced product. Each of the three lower bands shared the use of a cryptic splice donor located six nucleotides 5′ of the variant (AG_915_/GT), i.e., the c.921T variant located at the +5 position of a perfect consensus splice donor that is normally masked. In the lower band (band 4), this was the only difference in splicing from the expected wildtype band (band 1). For bands 2 and 3, they each shared a new cryptic splice acceptor site 135 nucleotides 5′ of the canonical exon 7 acceptor (i.e., within intron 6). This cryptic site conforms well to the splice acceptor consensus: 5′-(Y)11-13-N-(Y)-AG-3′ (where Y = a pyrimidine and N = any base): 5′-TCTCCCTTTATTGCAG|GA-3′; the vertical line marking the splice site [26]. The distinction between bands 2 and 3 is that band 3 contained only 44 nucleotides of the intronic sequence, whereas band 2 contained all 135 nucleotides up to the original start of exon 7. The inclusion of just part of the intron 6 sequence in band 3 is because of the utilization of a cryptic donor site (5′-AG|GTGGC-3′; vs. consensus: 5′-AG|GTRAG-3′) [26] that is then spliced to the canonical exon 7 acceptor site. The splicing pattern revealed by the sequencing of each product is schematically represented in Figure 2C. Importantly, each alternatively spliced mRNA is predicted to result in a frame shift and prematurely truncated IRF6 protein.

Somewhat surprisingly, alignment of the orthologous *IRF6* sequence from 100 vertebrate species found that all 59 mammalian species examined had the C allele, whereas 22 of 40 non-mammalian vertebrates (including marsupials, birds, reptiles and fishes) did not have a C at position c.921 or its equivalent position (see Appendix A). Note: the sequence over this region was not available for one vertebrate species. In all but two of these species, the C is substituted for a T (as seen in our family). Yet, notably, in seven of these species, the T substitution is associated with a corresponding substitution of the either G_915_ or G_916_ in the cryptic splice donor site (see Appendix A). Under the splicing models, this corresponding substitution is predicted to destroy the cryptic exonic splice site, thus countering the potential impact of the C>T change at position c.921.

### 3.4. The Splicing Pattern Is Affected by Ectopic Expression of Major Splice Regulatory Factors

In addition to the prediction that the synonymous variant created an ESS element, the Human Splice Finder tool also identified consensus binding sites for two major splice regulatory proteins, SRSF1 and SRSF2. Each of these factors can have both positive and negative effects on splicing depending on the context of their binding. We therefore decided to test whether the ectopic expression of either of these factors would impact the pattern or degree of alternatively spliced products seen with the variant minigene. As anticipated, the expression of either factor had no noticeable impact on the splicing of the wildtype minigene. With the variant minigene, however, the co-expression of SRSF1 improved the differential percentage between normal and abnormal splice products (mean difference = 32.93% (SD = 2.41) without SRSF1 vs. 25.53% (SD = 1.47) with SRSF1; two-tailed *t*-test: *p* = 0.000159). In contrast, the co-expression of SRSF2 exacerbated the difference between normal and abnormal splice products (mean difference = 43.57% (SD = 2.28) with SRSF2 vs. 32.93% without; *p* = 0.000030) (see Figure 3).

## 4. Discussion

Synonymous coding region changes, especially those away from utilized canonical splice sites, are often overlooked as primary causes of disease, as their impact has historically been difficult to predict and/or prove. However, in the recent medical literature, in large part due to improved bioinformatic algorithms and more sensitive analytical techniques, the number of examples where identified synonymous changes have been implicated in various human diseases has been growing. Diseases in which such changes have been shown to contribute include cancer [27,28,29,30,31,32,33,34,35,36], cystic fibrosis [37,38], Crohn’s [39,40], temporomandibular joint disorder [41], spinal muscular atrophy [42], congenital myasthenic syndrome [43], and osteoporosis [44]. Researchers have adopted a range of approaches to investigate the consequences of synonymous mutations, including population genetics models that demonstrate the negative selection of synonymous mutations at exonic splice enhancers (ESEs) [45,46,47,48], RNA sequencing and transcript expression analyses [27,29,30,31,32,34,35,36,37,39,43,44], and functional models that examine resulting protein and mRNA secondary structure and molecular interactions [29,31,35,37,38,39,41,44,49,50,51,52]. As a result, coding-independent mutations have been shown to disrupt gene regulation through a variety of mechanisms [28], including the disruption of splice sites and subsequent exon skipping [27,30,32,34,35,42,50,51,52], the creation of new splice sites [33], the loss of microRNA (miRNA) binding sites [29,39], protein misfolding [31,37,38], and mRNA secondary structure defects and instability [36,41].

Here, we describe the results of whole exome sequencing conducted on a multi-generational family of Honduran ancestry, in which multiple family members present with either isolated cleft palate or cleft lip with cleft palate. In what we believe is the first example for NS-CL/P, we identify a synonymous variant (c.921C>T; S307S) in a well-established cleft-causing gene, *IRF6*, that segregates with the orofacial clefting phenotype in all members that were available for testing. While the synonymous variant does not destroy a canonical splice site or create a new splice site, bioinformatic analysis suggested it created an exonic splice silencer (ESS) element. ESS elements are thought to bind heterogeneous nuclear ribonucleoproteins (hnRNPs) and generally function to inhibit neighboring splice site usage by interfering with components of the splicing complex. In doing so, they can promote the use of alternate splice sites. However, predicting the actual impact on splicing using bioinformatic algorithms is far from reliable [53]. Consequently, we chose to directly assess the impact of the variant using an established minigene splice assay [53]. This assay demonstrated a clear and significant disruption to the normal splicing of exon 7, detecting three principle aberrant splicing products. Sequencing of the individual products showed that each shared the unmasking of a cryptic splice site six nucleotides 5′ of the variant and suppression of the normal 3′ exon 7 splice site. Two of the products were also involved the concomitant utilization of a cryptic splice acceptor sequence located in intron 6, 135 nucleotides 5′ of the native exon 7 splice acceptor. One of these two products also utilized a cryptic donor site 44 nucleotides 3′ of the cryptic acceptor to join with the established exon 7 acceptor site. Each of the three abnormal splice products is predicted to result in a frame shift and premature truncation, and thus likely to result in nonsense-mediated decay (NMD) [54]. Together, the anomalous splicing impacted over 60% of the allelic splice products in the splicing assay. An important caveat to this result is that this assay is somewhat artificial and may not reflect the actual percentage of abnormal splicing of the endogenous *IRF6* locus. This would be most accurately determined using quantitative or real-time PCR using RNA from suitable tissue samples of the affected individuals [53], which unfortunately were not available. Nevertheless, these data do emphasize that the abnormal products represent a significant fraction of the total amount of *IRF6* transcript and that the loss of functional IRF6 protein in heterozygous individuals would also likely be significant.

ESS elements can also work in concert with exon splice enhancers (ESEs), as well as intronic silencers and enhancers (ISSs and ISEs) [53,55]. Indeed, some ESS elements can be juxtaposed with ESE elements to finely control splicing events, as in the case of exon 2 splicing of the HIV-1 *tat* gene [56]. Notably, binding sites for a number of serine/arginine splice factors (SRSF), which are splice regulatory proteins that bind ESEs, were also predicted to be centered over c.921 of *IRF6*. We therefore tested whether the ectopic expression of these SRSFs would mitigate the impact of the c.921T variant. Consistent with its known role in influencing splice site selection in a concentration-dependent manner [53], the ectopic expression of SRSF1 (ASF/SF2) led to a slightly improved ratio of normal to abnormal splice products. In contrast, SRSF2 (Sc35), which has a role in forming a bridge between the splicing components bound to the donor and acceptor sites, slightly exacerbated the abnormal splicing associated with the c.921T variant. These studies reinforce the complex nature of pre-mRNA splicing and highlight that much more work is needed to dissect out the impact of not only the splice regulatory factors but also the cis elements that govern their binding.

A number of synonymous variants have previously been linked to orofacial clefting. Letra et al. identified a c.471C>T (rs8061351; p.Pro157Pro) variant in the *CRISPLD2* gene through family-based association and found that the C allele was significantly overrepresented in CL/P cases [57]. In silico analysis predicted the alteration of an ESE motif, and also created a putative binding site for transcription factor AP-2α, an important regulator of craniofacial development. However, no functional studies were performed to verify either possibility. In a case–control association study of NS-CL/P in an Indian population, Kumari et al. found an association with a synonymous c.348C>T variant (rs34165410; p.Gly116Gly) in the *MSX1* gene [58]. In silico modeling of the NS-CL/P-associated variant suggested the variant may create a new splice site and/or affect expression through decreased codon usage. Functional testing of the variant in a minigene splicing assay failed to demonstrate an impact on splicing. More recently, Gaczkowska et al. identified nucleotide variants in the *PAX7* gene that were overrepresented in NS-CL/P through a GWAS of a Polish population [59]. In a subsequent evaluation of the *PAX7* gene in isolated NS-CL/P cases, they identified a synonymous substitution (c.87G>A, p.Val29Val) in a single patient. In silico modeling predicted the disruption of a potential ESE motif in the NS-CL/P-associated variant but this was not functionally tested. Therefore, we believe our case represents the first such instance of a demonstrable functional impact of a synonymous variant in NS-CL/P. Functionally impactful synonymous variants, including those affecting exonic splice regulatory elements, have been previously described in a number of disorders, including cancer [30,35,60,61], neurofibromatosis type 1 [62], severe combined immunodeficiency [63,64], familial dilated cardiomyopathy [65], and hemophilia B [66]. Given the broad range of phenotypes and disease severity in NS-CL/P, we believe that synonymous variants should not be quickly dismissed but rather more actively pursued, as they could underlie important variation in gene expression throughout the development of the face. Our studies also highlight the necessity to functionally test the impact of any such variants, whether non-synonymous or synonymous, on splicing, as highlighted by studies on *BRCA1* [67].

The family described in this study lives in a rural part of Honduras, and we have been unable to make contact with them, despite attempts over the last five years, to inquire whether we might be able to follow up the study. While we would have preferred to collect saliva samples from all family members presenting to the initial Operación Sonrisa screening, several were minors accompanied by non-guardian family members, and thus could not be properly consented. The lack of sequencing data from additional family members is a limitation to this study. Nevertheless, we were able to confirm the variant in all affected individuals and obligate carriers on both sides of the pedigree for whom samples were available. Furthermore, sequencing and confirmation of the variant in unaffected individuals, while interesting, would not necessarily contradict our findings, since the phenotype is incompletely penetrant.

Practical difficulties notwithstanding, if the family members can be re-assembled and consented for further research, direct analysis of their *IRF6* transcript sequences and expression levels would provide valuable validation of our in vitro findings. In the future, it may be possible to create an animal model of the variant described in this study. While animal model studies are a crucial step toward discovering underlying mechanisms of disease treatments, few animal models of synonymous change-driven disease have been reported. In a number of studies of spinal muscular atrophy animal models, in which a synonymous mutation was associated with the loss of an ESE and the antagonistic gain of a silencer, the in vivo administration of antisense oligonucleotides that mask the silencer vastly improved symptoms [42,50,51,52]. This study highlights the future potential of similar approaches in utero, in conjunction with prenatal genetic screening, for ensuring proper development and preventing synonymous change-driven NS-CL/P.

## Figures and Tables

**Figure 1 genes-11-00903-f001:**
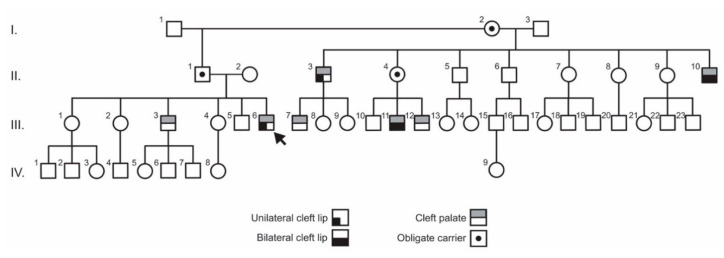
Pedigree of Honduran family with non-syndromic cleft lip and/or palate (NS-CL/P). Pedigree shows inheritance from individual I-2 through both sides of the pedigree: II-1, III-3, III-6 & II-3, III-7, II-4, III-11, III-12, and II-10. Arrow marks the proband (individual III-6).

**Figure 2 genes-11-00903-f002:**
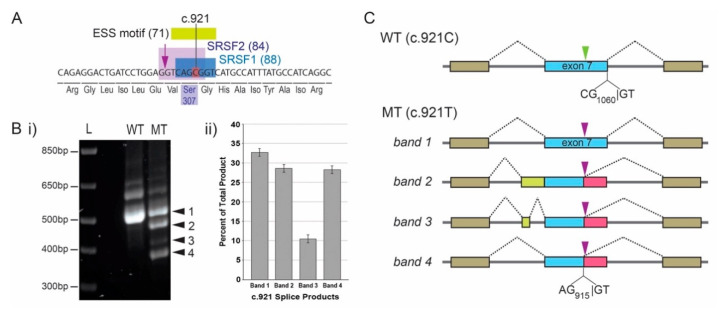
The synonymous *IRF6* c.921C>T variant impacts splicing. (**A**) In silico analysis predicted the creation of an exonic splice silencer (ESS) element (score: 71) and enhanced putative recognition sites for the splicing factors, including SRSF1 and SRSF2. The scores are indicated in brackets. (**Bi**) In vitro minigene splice assay detected multiple abnormal splice products (bands 2–4) in addition to the normal-sized product (band 1). Note: the slower migrating bands were seen with both the wildtype and mutant and are thought to represent pre-spliced forms, as they were absent from control PCRs using the RNA (without reverse transcriptase) as a template. (**Bii**) Quantification of splice products revealed that abnormal products constituted ~67% of allelic transcripts. (**C**) Sequencing of purified splice products resolved the nature of the underlying splice events. The arrowheads depict the location of nucleotide position c.921: C = green; T = purple. The variant unmasks the cryptic exonic splice site at position c.915, as well as a site within intron 6 (135 nucleotide 5′ of the canonical 5′ splice acceptor of exon 7). The canonical acceptor site is used differentially in bands 2 and 3.

**Figure 3 genes-11-00903-f003:**
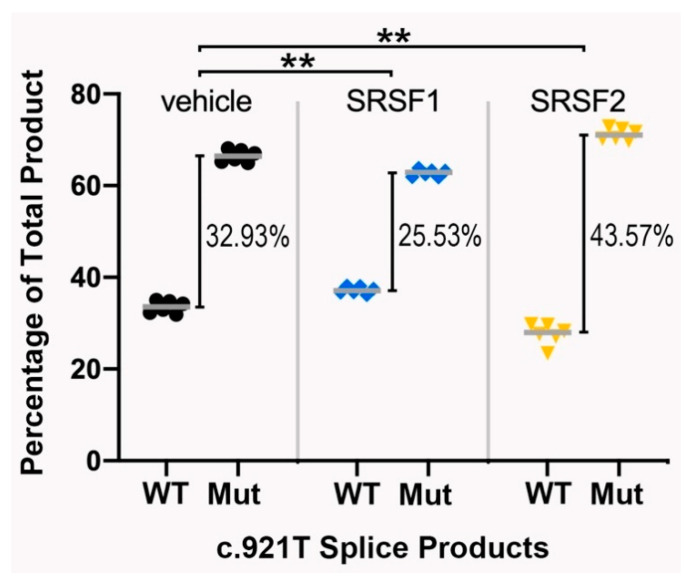
The splice regulatory factors impact the extent of abnormal splicing caused by the synonymous *IRF6* c.921C>T variant. The total amount of PCR splice products (WT: band 1, Mut: bands 2–4, as per Figure 2B) were quantified and expressed as a percentage of the total. Six replicates were performed. Statistical analysis was carried out using two-tailed Student’s *t*-tests; ** *p* < 0.05. Actual *p*-values from the tests are presented within the main text.

**Table 1 genes-11-00903-t001:** Shared variants identified by exome sequencing. From the original list of 411 shared variants, further filtering based on DOMINO class consistent with a dominant mode of inheritance and minor allele frequency of less than 1 × 10^−4^. (VLD = very likely dominant; LD = likely dominant; EDR = either dominant or recessive). Note: Allele count and minor allele frequency (MAF) were determined from the gnomAD database (June, 2020).

Genomic Coordinate	Gene Name	Location	Type	Protein Change	CADD Scaled	DOMINO Class (Probability)	gnomAD Allele Count (MAF)	Associated Disease
chr8:g.13251073G>C	DLC1	Coding	Missense	p.Pro435Ala	23.4	LD (0.706)	0	Nephrosis
chr1:g.155838558G>A	SYT11	Coding	Synonymous	p.Arg279Arg	11.89	EDR (0.478)	0	
chr1:g.114940211G>T	TRIM33	3’UTR	SNP		0.742	VLD (0.997)	0	
chr3:g.149684103G>A	PFN2	3’UTR	SNP		16.21	VLD (0.948)	0	
chr1:g.16262494_16262499dupTGTCCC	SPEN	Coding	Duplication	p.Val3254_Pro3255dup		VLD (1)	0	
chr5:g.149922536G>A	NDST1	Intronic	Splice		14.37	VLD (0.964)	0	Mental retardation
chr11:g.156907027_156907033delCCTGCTT	ARHGEF11	Intronic	Deletion			LD (0.794)	0	
chr7:g.137597556G>C	CREB3L2	Intronic	Splice		5.642	LD (0.616)	0	Fibromyxoid sarcoma
chr8:g.38947723delT	ADAM9	Intronic	Deletion			EDR (0.584)	0	Cone-rod dystrophy
chr18:g.59936640dupC	KIAA1468	Intronic	Splice			VLD (0.959)	0	
chr12:g.64882264delT	TBK1	Intronic	Splice			VLD (0.882)	0	Frontotemporal dementia/ALS
chr9:g.73458046dupA	TRPM3	Intronic	Splice			EDR (0.585)	0	Intellectual disability/epilepsy
chr19:g.10798342G>A	ILF3	Coding	Missense	p.Gly798Arg	24.2	LD (0.677)	2 / 247274 (8.09 × 10^−6^)	
chr1:g.209963979G>A	IRF6	Coding	Synonymous	p.Ser307Ser	14.48	LD (0.733)	3 / 251352 (1.19 × 10^−5^)	Orofacial clefting
chr8:g.66647047A>G	PDE7A	Coding	Synonymous	p.Cys226Cys	12.85	EDR (0.407)	3 / 250766 (1.20 × 10^−5^)	
chr12:g.6094200G>A	VWF	Intronic	SNP		5.659	LD (0.617)	6 / 251458 (2.39 × 10^−5^)	von Willebrand disease
chr2:g.240085511C>T	HDAC4	Coding	Missense	p.Arg200His	25	VLD (0.999)	7 / 251428 (2.78 × 10^−5^)	Brachydactyly, mental retardation
chr8:g.26484831G>A	DPYSL2	Intronic	SNP		0.659	VLD (0.995)	9 / 282698 (3.18 × 10^−5^)	
chr1:g.16268440_16268443delTAAA	SPEN/ZBTB17	3’UTR	Deletion			VLD (1)	7 / 218068 (3.21 × 10^−5^)	
chr7:g.98555591G>T	TRRAP	Intronic	SNP		0.096	VLD (1)	10 / 248972 (4.02 × 10^−5^)	Developmental delay/autism, dysmorphic facies
chr1:g.150551341G>A	MCL1	Coding	Synonymous	p.Arg222Arg	12.07	VLD (0.997)	13 / 281774 (4.61 × 10^−5^)	
chr17:g.78113791C>A	EIF4A3	Intronic	SNP		0.049	VLD (0.999)	12 / 244608 (4.91 × 10^−5^)	Robin sequence, cleft mandible, limb anomalies
chr1:g.154527884G>A	UBE2Q1	Intronic	SNP		4.602	LD (0.727)	13 / 251300 (5.17 × 10^−5^)	
chr1:g.155948197T>G	ARHGEF2	Coding	Missense	p.Thr8Pro	22.4	VLD (0.916)	14 / 167764 (8.35 × 10^−5^)	Neurodevelopmental disorder, brain malformations
chr10:g.33196031T>C	ITGB1	coding	Missense	p.Lys791Arg	22.6	VLD (0.996)	26 / 279598 (9.30 × 10^−5^)	
chr12:g.52911558C>T	KRT5	intronic	SNP		5.636	EDR (0.589)	28 / 282824 (9.90 × 10^−5^)	epidermolysis bullosa
chr17:g.72759561G>A	SLC9A3R1	coding	Missense	p.Arg220Lys	14.76	VLD (0.865)	25 / 250294 (9.99 × 10^−5^)	Nephrolithiasis/osteoporosis, hypophosphatemia

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
