# Peer review of "A Synonymous Exonic Splice Silencer Variant in IRF6 as a Novel and Cryptic Cause of Non-Syndromic Cleft Lip and Palate"

_genes, 2020, doi:10.3390/genes11080903_

Round 1
Reviewer 1 Report
The authors carried out whole exome sequencing to identify a mutation in the IRF-6 gene which segregated with cleft in a family with several cleft cases.
Whilst it is interesting study, the mutation is very rare (perhaps family specific) so the paper is unlikely to have wider applicability, but nevertheless adds to the evidence on cleft variants.
I do have some concerns with the study:
1) The authors state that a dominant model with incomplete penetrance is the most likely model, but the segregation in the family could also suggest an x-linked recessive model, particularly if there is some consanguinity within the pedigree. Please could the authors discuss why they ruled this out.
2) The authors ruled out several other variants in favour of the IRF-6 variant, some of which were strong candidates, I would like to see more discussion as to why they were ruled out.
3) Very few family members were genotyped which weakens the findings from this study, I understand that it was not possible to genotype other individuals but I think this limitation should be discussed further.
Author Response
Response to Reviewer 1 Comments
Point 1: The authors state that a dominant model with incomplete penetrance is the most likely model, but the segregation in the family could also suggest an x-linked recessive model, particularly if there is some consanguinity within the pedigree. Please could the authors discuss why they ruled this out.
Response 1: At the start of any family study we consider all modes of inheritance. However, the instances of male-to-male transmission, as from individuals II-1 to III-3 and III-6 and from II-3 to III-7 (see Figure 1 in Section 3.1, lines 175-180), rule out this explanation.
Point 2: The authors ruled out several other variants in favour of the IRF-6 variant, some of which were strong candidates, I would like to see more discussion as to why they were ruled out.
Response 2: As requested, we have now included (in Section 3.2, lines 190-218) more detail on our process of consideration of the variants in Table 1, including why other variants were excluded or deprioritized, and the IRF6 variant considered the most likely cause.
Point 3: Very few family members were genotyped which weakens the findings from this study, I understand that it was not possible to genotype other individuals but I think this limitation should be discussed further.
Response 3: We agree that the lack of sequencing data from additional family members is a limitation of this study and have added discussion of this point (lines 486-502), addressing difficulties in re-establishing contact with the family and consenting minors during initial presentation. We also noted that we were able, at least, to establish segregation of the studied variant in all affected individuals and obligate carriers who were sequenced and that sequencing of additional unaffected individuals, while valuable, would not prove or disprove our hypothesis due to incomplete penetrance.

Reviewer 2 Report
Summary:
The authors report the finding of a synonymous variant, located in middle of exon 7 of IRF6, shared among several members of a family of Honduran ancestry presenting with variable severity of forms of cleft-lip and/or palate (CL/P). IRF6 is a prominent CL/P gene, which can be affected by missense, nonsense, splice site and regulatory region variants. Here, the authors describe a synonymous variant in IRF6 that results in abnormal splicing events and likely truncating the protein. Their analysis suggests the variant to impact an exonic splice silencer element and unmask a cryptic splice site. Interestingly, the authors state that major splice regulatory factors may be involved in the abnormal splicing events. They further highlight that this is the first study to identify a synonymous variant as a likely cause of non-syndromic CL/P and point to care to be taken to include synonymous variants in future analysis and interpretations.
Results:
The authors find through exome sequencing and variant analysis that IRF6 contains a synonymous change located in middle of exon 7 of affected family members and is the likely culprit among 27 gene variants in these members. Indeed, IRF6 is the most prominent CL/P gene and Sanger sequencing further validated the presence of the IRF6 variant.
After this characterization and selection of IRF6 as their most likely candidate, the authors continue with functional testing of this IRF6 variant. Here, they focus on potential gene regulation elements that may be disrupted due to this synonymous mutation. In silico analysis suggest that a new exonic splice silencer (ESS) element is created due to the variant which they verify to result in disrupting mRNA splicing/alternative splicing using an in vitro mini-gene splice assay. Importantly, they find that each alternatively splicing event is predicted to result in a truncating frameshift of IRF6. The authors also confirm using direct sequencing that alternative splicing events make use of cryptic splice acceptor site, either an unmasked site or a new cryptic splice acceptor site.
The authors then make an interesting observation that in some species with the C>T substitution the cryptic exonic slice site may be destroyed, potentially countering a deleterious effect. Unfortunately, that data is not presented. It would be of interest to show the full data behind this analysis and which species were included.
Finally, the authors provide evidence using regular PCR that two splice regulatory factors have differential effect on alternative splicing pattern between the WT variant and the mutant variant. While the direction of effect is likely to exist, it is debatable whether end-point PCR is suitable to provide such precise estimations of the size of the effect. It would be necessary that the authors provide arguments why in this case end-point PCR is sufficient to quantify four mRNA splice variants after their cDNA conversion or perform a more quantitative assay. In addition, DNA gel electrophoresis quantification suffers from the limitation that the signal is dependent on the size of the DNA fragment. It does not become clear of the authors have adjusted their calculations for that, which would be necessary. The differential regulation by SRSF1 and SRSF2 would be an interesting finding, especially considering the variable expressivity and male-dominance found of CL/P in this family.
Opinion:
The authors report a novel variant in a prominent CL/P gene. The finding is of importance because it’s the first report of a synonymous variant as a likely cause of NS-CL/P. Therefore communication, through this manuscript, to laboratories and clinical geneticist that synonymous variants should be included for further analysis as potential causes of NS-CL/P is key. A further implication of SRSF1 and SRSF2 is of interest for future studies to further understand dysregulation of mRNA splicing as a cause for CL/P. However, the authors first need to address the two methodological points (described as Major point 1 & 2) to further strengthen the presented evidence of these interesting findings.
Major point:
- Figure 3 & Line 258, section 3.4 and further: The suitability and validity of end-point PCR as quantitative measure to determine reported effects should be made evident. As made clear above by this reviewer, end-point PCR is generally not considered a quantitative method and at a minimum the authors should provide reasoning why it is sufficiently useful here or provide additional experimental evidence (using either a control with spiked/known-quantities of the splice variants in their current assay or perform an alternative quantitative (PCR) method).
- In addition, quantification from a DNA electrophoresis gel (Figure 2B) should consider the size difference of the DNA fragments. For instance, equimolar amounts of two fragments: a fragment A and a fragment B—half the size of A—will result in approximately 50% the intensity since A has two times more total DNA compared with B. Since in this case there is up to ~25% difference between the WT and MT fragments (Figure Bi), any quantification without correcting for the size would underestimate the MT amounts. It is not clear from the Methods section that any correction has been done. This analysis and any potential correction would impact Figure 2B and Figure 3. The authors should state whether a correction for size was performed or perform a correction. The necessity of this point depends on the outcome om Major point #1.
Minor points:
It would be preferable if the authors discuss the differential regulation by SRSF1 and SRSF2, and to what extent it may affect causes of CL/P. Of specific interest is to what extent the variable expressivity and male-dominance can be explained by the differential regulation by SRSF1 and SRSF2, if such a case can be made.
Textual:
Line 72: add the word “studies” after “NS-CL/P in multiple family-based” for clarity
Line 252: “C)” in the legend is “Bii)” in the figure. Correct so that Figure and legend annotation are same.
Line 253: “D)”is “C)”.
Author Response
Point 1: The authors also confirm using direct sequencing that alternative splicing events make use of cryptic splice acceptor site, either an unmasked site or a new cryptic splice acceptor site. The authors then make an interesting observation that in some species with the C>T substitution the cryptic exonic slice site may be destroyed, potentially countering a deleterious effect. Unfortunately, that data is not presented. It would be of interest to show the full data behind this analysis and which species were included.
Response 1: As requested, we have submitted a supplementary figure with the full species data and included a reference to it in the text (line 275).
Point 2: Figure 3 & Line 258, section 3.4 and further: The suitability and validity of end-point PCR as quantitative measure to determine reported effects should be made evident. As made clear above by this reviewer, end-point PCR is generally not considered a quantitative method and at a minimum the authors should provide reasoning why it is sufficiently useful here or provide additional experimental evidence (using either a control with spiked/known-quantities of the splice variants in their current assay or perform an alternative quantitative (PCR) method).
Response 2: The reviewer is correct in that, typically, end-point PCR is not a suitable method for quantification. However, we reasoned that the splice assay itself is not quantifying actual endogenous mRNA splicing but rather from a synthetic construct which is transfected into cells in culture. Although we show consistent amounts of each product between experiments, transfection efficacy of cells in this assay often varies (for the cell line used we see transfection rates of 60-85%). Regardless of the level of cell transfection, the proportions of each splice product remain similar. Even so, quantification by qPCR, for example, would under normal circumstances provide a more accurate quantitation of the relative amounts of product. However, this is also complicated by the fact that qPCR requires each individual band to be amplified separately (unique primer pair) to enable quantification. Bands 1 and 4 differ from each other only by the absence of internal sequence in band 4. Bands 2 and 3 differ in a similar manner – band 3 missing some sequence seen in band 2. Given the artificial nature of the assay, we felt quantification of a standard PCR would suffice as a guide. But we appreciate it has some limitations. Of course, it would be most valuable if this analysis could have been performed on cells isolated from one of the affected member of the family, but such samples were not available. Nevertheless, we feel that providing this data does emphasize that the products represent a significant fraction of the total.
Point 3: In addition, quantification from a DNA electrophoresis gel (Figure 2B) should consider the size difference of the DNA fragments. For instance, equimolar amounts of two fragments: a fragment A and a fragment B—half the size of A—will result in approximately 50% the intensity since A has two times more total DNA compared with B. Since in this case there is up to ~25% difference between the WT and MT fragments (Figure Bi), any quantification without correcting for the size would underestimate the MT amounts. It is not clear from the Methods section that any correction has been done. This analysis and any potential correction would impact Figure 2B and Figure 3. The authors should state whether a correction for size was performed or perform a correction. The necessity of this point depends on the outcome om Major point #1.
Response 3: We acknowledge that we did not adjust the quantified data to account for this and so we have now done this in the revised manuscript, updating both Figures 2Bii and 3. While this does not change the interpretation of the findings in any way, this accommodation does increase the calculated proportion of abnormal splice products relative to wildtype (enhancing the support for pathogenicity) as well as the magnitude of effect of co-expression of the splice regulatory factors.
Point 4: It would be preferable if the authors discuss the differential regulation by SRSF1 and SRSF2, and to what extent it may affect causes of CL/P. Of specific interest is to what extent the variable expressivity and male-dominance can be explained by the differential regulation by SRSF1 and SRSF2, if such a case can be made.
Response 4: It is certainly an interesting concept raised by the reviewer. In many cancers, aberrant splicing is a notable feature and in particular, in myelodysplastic syndrome, there has been reported a higher degree of global aberrant splicing in males (Yang et al, 2018). Mutations in ESRP2 in humans and ESRP1/2 in mice have been shown to cause facial clefts, although both splicing factors are epithelial-specific in the function and no sex-bias has been reported. While it is conceivable, to the best of our knowledge there is no evidence suggesting the regulation of splicing contributed to the known male sex bias in CLP. And neither IRF6 nor the general splice regulatory factors, SRSF1 and SRSF2, are encoded by X-linked genes. At this stage it would be too speculative to mention such a role.
Textual:
Line 252: “C)” in the legend is “Bii)” in the figure. Correct so that Figure and legend annotation are same.
Line 253: “D)”is “C)”.
We have corrected the legend to reflect the figure.

Round 2
Reviewer 2 Report
Reviewer response to Response 1: Accepted.
Reviewer response to Response 2: This reviewer understands that this assay does not quantify endogenous mRNA splicing from subjects, rather the splicing products from a synthetic construct of which the ratios of products is independent of transfection efficiency and that qPCR for these products is complicated for the given reasons. The reviewer is happy to see the authors’ acknowledgements that this assay has limitations and will accept this point after these limitations have been clarified in the manuscript's text and that “this data does emphasize that the products represent a significant fraction of the total” rather then emphasis on the specific amounts.
In addition, a comment should be made that the quantification goes beyond the “established minigene assay” (line 332 and elsewhere) from citation # 53 (Abramowicz, A.; Gos, M. Splicing mutations in human genetic disorders: examples, detection, and confirmation. J Appl Genet 2018, 59, 253-268, doi:10.1007/s13353-018-0444-7.), which states that “These techniques are limited because they do not assess the relative level of transcript isoforms. The most valuable method to resolve this problem is real-time PCR (RT-PCR) or quantitative PCR (qPCR) which enables to measure the quantity of each mutated transcript and compare it to the level of the nonmutated one.”
Reviewer response to Response 3: Accepted after adding text on calculation and adjusting for size in the Methods section.
Reviewer response to Response 4: Accepted.
Further textual points:
Line 199: gene = genes
Line 232 vs 332 and elsewhere: consistency in use of word: mini-gene vs minigene
